# Unintentional Tobacco Smoke Exposure in Children

**DOI:** 10.3390/ijerph19127076

**Published:** 2022-06-09

**Authors:** Caseng Zhang, Kaden Lam, Patrick Hicks, Matt Hicks, Lesley Brennan, Irena Buka, Anne Hicks

**Affiliations:** 1Faculty of Health Sciences, McMaster University, Hamilton, ON L8S 3L8, Canada; zhanc106@mcmaster.ca; 2Cumming School of Medicine, University of Calgary, Calgary, AB T2N 4N1, Canada; kaden.lam@ucalgary.ca; 3Faculty of Kinesiology, Sport, and Recreation, University of Alberta, Edmonton, AB T6G 2H9, Canada; pmhicks@ualberta.ca; 4Department of Pediatrics, University of Alberta, Edmonton, AB T6G 2H9, Canada; mhicks1@ualberta.ca (M.H.); lbrennan@ualberta.ca (L.B.); iwbuka@gmail.com (I.B.)

**Keywords:** secondhand smoke, cotinine, child

## Abstract

Secondhand smoke (SHS) exposure increases the prevalence and severity of sinopulmonary diseases in children. The primary source of SHS exposure in children is through adults who live in the same house; however, the level of exposure may vary based on the adult smoking habits at home. This prospective cross-sectional study in Alberta, Canada, investigated the relationship between self-reported caregiver smoking, location, outdoor temperature and children’s’ urine cotinine: creatinine ratio (CCR), a marker of nicotine metabolism. Participants aged 0–9 were recruited from the Child Health Clinics at the Misericordia Community Hospital in Edmonton, Alberta, from 8 January to 24 February 2016 and 30 June to 18 August 2016. Participant CCR levels were compared to caregiver-reported smoking location and environmental factors such as temperature and season. Of the 233 participants who reported smoking status, 21% reported smoking, in keeping with local smoking rates. More participants smoked indoors during the winter than the summer; however, some families limited indoor smoking to a garage. Of the 133 parent–child dyads who provided smoking information and a child urine sample, 18 had an elevated cotinine:creatinine ratio, suggestive of significant tobacco smoke exposure, 15 of whom were from homes that reported smoking. Age < 1 year and number of cigarettes smoked in the home weekly were risks for significant exposure while season, outdoor temperature and smoking location in the home did not reach significance. Smokers should be counseled to protect children, particularly infants, from exposure by limiting the number of cigarettes smoked and isolating smoking to outside the home. Segregated areas such as a garage may provide a useful harm mitigation strategy for indoor smokers, provided the garage does not share ventilation or is not in close proximity to high-traffic areas of the home.

## 1. Introduction

Secondhand smoke (SHS) refers to passive exposure to smoke, typically from tobacco product use [1,2]. In children, SHS exposure increases the incidence, prevalence and severity of sinopulmonary diseases including asthma, bronchiolitis, ear infections and pneumonia as well as sudden infant death syndrome [1,3]. Up to 40–70% of children are exposed to SHS globally, and approximately 14% of children are exposed to SHS in Canada [4,5]. Potential exposures to thirdhand smoke (THS), the residual chemicals left on surfaces following smoking, present another route to indirect pediatric exposure [6].

Children are most likely to be exposed to SHS by adults who live in their homes, primarily on a balcony or in the yard. The child’s bedroom is the location of the least exposure in homes with SHS [7]. While few studies focus on risk at the level of locations in the home, more studies have identified that outdoor exposures at home are more common than indoor exposure [7,8]. Multi-unit homes (MUHs) and homes near outdoor smoking areas are also responsible for exposure through shared air spaces, ventilation systems, windows, elevators, hallways, leaks in walls and open windows [7,9]. Although outdoor smoking is a significant source of home SHS exposure, there is limited exploration of the influence of temperature or season on SHS exposure in children; in places where extreme hot and cold temperatures are common, smokers may change their smoking location from outdoors to indoors.

Cotinine, a primary metabolite of nicotine, is detectable in the serum, urine, saliva and hair of smokers and individuals exposed to SHS, with a positive association between the amount of cotinine and total exposure [4,7,10]. It is a useful biomarker for tobacco exposure as it has a longer plasma half-life than nicotine that reflects recent exposure [3,11]. Urine levels in nonsmokers exposed to SHS range from <1 ng/mL for low-level exposure to 10 ng/mL for higher-level exposure [10,12]; active smokers usually have >10 ng/mL, up to >500 ng/mL [11]. The urine cotinine:creatinine ratio (CCR) can be used to address potential discrepancies due to inconsistent urine concentrations [13].

This project evaluated the association between outdoor temperature, caregiver self-reported smoking and urine cotinine in children, including a descriptive characterization of home sources of SHS and the effect of environmental factors such as temperature or seasonality on smoking behaviors.

## 2. Materials and Methods

### 2.1. Study Design

This prospective cross-sectional study evaluated the environmental tobacco smoke exposure in children through parental self-report and child urine CCR. Participants were recruited from the Child Health Clinics at the Misericordia Community Hospital in Edmonton, Alberta, from 8 January to 24 February 2016 and 30 June to 18 August 2016. Inclusion criteria: children aged 0–9 years, to avoid collecting samples from actively smoking children; approximately 13% of Canadian children in grades 6 and up have tried a cigarette at least once [14]. Exclusion criteria: parent or caregiver unable to complete informed consent. This project received approval from the University of Alberta Human Ethics Research Board Pro00060475.

### 2.2. Urine Collection and Analysis

For older children, urine was collected in laboratory specimen jars. For infants, cotton balls in the infant’s diaper were used to collect urine and then centrifuged to collect the urine from the cotton balls. Urine cotinine was measured by ELISA (CalBiotech Cotinine Direct ELISA Kit). Creatinine was measured using a colorimetric assay (Cayman Chemical Co. Ann Arbor, MI, USA) as per manufacturer’s instructions. A cutoff of 30 ng/mg cotinine/creatinine or above was used as an indicator of passive smoking [15,16].

### 2.3. Questionnaires

Questionnaires collected the child’s age and household smoking information from the last month, including the number of smokers and frequency of tobacco use in different indoor and outdoor areas of the home, garage and primary vehicle. The location of children during smoking was not recorded. They were completed by parents/guardians living in the same household as the child at the time of study recruitment.

### 2.4. Season and Temperature

Winter recruitment was carried out between 8 January and 24 February 2016, while summer recruitment was carried out between 30 June and 18 August 2016. The midday outdoor Edmonton temperature was recorded daily by the recruiter.

### 2.5. Statistical Analyses

Statistical analyses were performed using IBM^®^ SPSS^®^ Statistics for Windows and Mac, Version 28 (IBM Corp., Armonk, NY, USA) and Microsoft Excel™ (Microsoft Corporation, Redmond, WA, USA). Descriptive statistics were presented as counts for nominal variables. CCR ratio was analyzed against reported SHS exposure using the chi-squared test or Fisher’s exact test according to sample size. All variables included a bivariate analysis. A significance level of 0.05 was set for all statistical tests. Based on bivariate analysis, variables were considered in multivariate logistic regression models in an iterative backward and forwards stepwise selection process. Variables that included multiple assessments of smoking were not included together in the logistic regression to prevent covariance.

## 3. Results

### 3.1. Participants

A total of 241 parent–child dyads enrolled; 233 completed the questionnaire, 141 provided urine samples and 133 completed both; and 8 were excluded for not completing the questionnaire and 100 from the complete analyses due to lack of a urine sample.

Of the 233 participating dyads who completed the questionnaire, 173 (74%) reported that there was no tobacco smoke exposure in the home; 21 (9%) limited smoking to outdoors, 19 (8%) to the garage or outdoors but not in the main part of the home, and 20 (9%) reported smoking inside the home. The 209 (90%) participants who completed questionnaires reported residing in single detached homes. Smoke outside of the home at least once a week was reported by 29 (12%) of participants, from locations including public areas, the homes of relatives, neighbors and friends. There were 100 participants in the winter, with 27 smokers (27%) and 133 in summer, with 23 smokers (17.3%) (Table 1). Of the children in homes that reported smoking, 11/46 were <1 year old (23.9%), 24/105 were 1–4 years (22.9%), and 11/69 were 5–9 years (15.9%). The average number of reported smokers per household in smoking homes was 2.3 for children <1 year, 1.5 for children 1–4 years and 2.9 in homes with children 5–9 years.

Of the 133 participants for whom both urine samples and questionnaires were available, 30 were <1 year, 49 were 1–4 and 52 were 5–9; 2 had incomplete age data (Table 2); 62 were collected during the summer, with 17/62 (27.4%) reporting smoking and 71 during the winter, with 20/71 (28.2%) reporting smoking (Table 3). The 115 (86%) participants who provided urine samples and completed questionnaires resided in single-family housing.

For the 18 participants whose urine sample met the CCR cutoff for tobacco smoke exposure, 15 lived in homes with reported smoking. The range of smokers per home that reported smoking was 1 to 6 with a mean of 1.8. Six adult smokers in a home were reported by three families, and eight families reported smoking indoors. Eleven CCR-positive participants were under the age of 1 year (11/30 [36.7%]) (Table 2), four were ages 1–4 (4/49 [8.16%]) and three ages 5–9. Of the parent–child dyads whose children provided a urine sample and parents completed the questionnaire, 35 reported smoking in at least one environment; for children under the age of 1 year, 37% lived in homes with self-reported smokers (11/30), whereas only 27% of children aged 1–4 (13/49) and 17% of children aged 5–9 (9/52) lived in homes with smokers.

### 3.2. Environmental Factors

For the 233 participants who completed questionnaires, temperature data were missing for 15 participants; the other 218 were distributed within temperature quartiles based on the highest and lowest recorded temperatures during the recruitment period: −15 °C–−5 °C (49/218 [22.5%]); −4 °C–5 °C (38/218 [17.5%]); 6 °C–16 °C (19/218 [8.6%]); and 17 °C–27 °C (112/218 [51.4%]) (Appendix A). Of those, 14/49 (28.6%) and 5/38 (13.2%) reported smoking at home in the lowest two quartiles, respectively, with an average of 2.3 and 3.4 smokers per home; in the third quartile, 3/19 (15.8%) reported smoking, with an average of 2.7 smokers per household, and in the highest temperature quartile 21/112 (18.8%) reported smoking, with an average of 1.3 smokers per household. A significant number of winter participants did not report smoking location (Table 1); for those who did, the proportion of indoor smokers was similar to those who smoked outdoors only or limited indoor smoking to a garage, although trends did not reach significance. Conversely, in the summer, more participants reported the location of smoking, but more than half indicated limiting indoor smoking to the garage, with trends not reaching significance (Table 1).

For the 133 participant dyads who completed questionnaires and provided a child urine sample, 7/17 who reported smoking in the winter did not report the location. Smoking was limited to outdoors or the garage for four families, while six reported indoor smoking, with no trend reaching significance. Participants in the summer were more likely to report smoking location (80%), and outdoor smoking was more prevalent (8/20; 40%) than indoor smoking (2/20; 10%), with 6/20 (30%) smoking in the garage. Temperature quartiles were analyzed for 128 samples (5 did not include temperature data): −15 °C–−5 °C (31/128 [24.2%]); −4 °C–5 °C (25/128 [19.5%]); 6 °C–16 °C (12/128 [9.38%]); and 17 °C–27 °C (60/128 [46.9%]) (Appendix A). Four participants met the CCR cutoff in the lowest two temperature quartiles (4/32 [12.9%] vs. 4/25 [16.0%]); two in the third quartile (2/12 [16.7%]) and eight in the highest temperature quartile (8/60 [13.3%]).

Seasonality (Table 1 and Table 2), outdoor temperature (Appendix A) and child health conditions had no influence on being above the CCR cutoff. Several smoking location factors demonstrated significant associations with meeting the CCR cutoff in bivariate analyses (Table 4). However, when factors were included in multivariate logistic regression modelling, the only factor that reached significance was the number of cigarettes smoked in the household [OR: 16.0875, 95% CI (5.0495–51.25352)]. This variable also demonstrated a dose response based on the number of cigarettes smoked in the household. Smoking one to ten cigarettes a week showed an OR of 9.9 (95% CI 2.761–35.49), while smoking more than ten cigarettes a week had an OR of 59.4 (95% CI 9.478–372.23).

## 4. Discussion

SHS exposure increases the incidence, prevalence and severity of childhood sinopulmonary diseases, including asthma and sudden infant death [1,3]. For many smokers, cessation is not an option; harm mitigation strategies to decrease children’s exposure may improve their health. Outdoors or outdoors and garage only were the most popular smoking locations in our study; for participants who provided a urine sample as well as completing questionnaires, outdoors was the most common. This is in keeping with a similar study among middle school students in northern Thailand [9], which noted that outdoor locations made up 53.5% of SHS exposures. Smoking outside may be more convenient in Thailand than in Canada because of its warmer climate. Despite this difference in climate, exposure to SHS is also higher outdoors than indoors in Canada [8]. In a recent Canadian study, it was reported that among the 12% of households that reported smoking at home after a child’s birth, 11% smoked outside of the house, 1.6% smoked near a window or in the garage, and only 0.5% smoked inside the home [8].

While outdoors was reported to be the most common location of smoking and is associated with higher exposure to SHS, THS should also be considered, especially in households with indoor smokers. In a 2004 study addressing THS, urine cotinine levels were significantly higher in infants who lived in households with indoor smokers compared to households with outdoor smokers [17]. There is a lack of public awareness and understanding of THS exposure, with 85% of tobacco smoke not being odorous or visible, leading to a gap in knowledge among the public [8].

Although there were differences between CCR and home smoking location in our study and reported differences in smoking location based on the season or temperature, seasonality was not linked with a significant difference in urine CCR in child participants, and smoking location did not make a significant difference in children that met a CCR cutoff suggestive of exposure, likely due to an insufficient number of participants for significant differences to be observed. We found higher CCR ratios in the summer, and at warmer temperatures, than in the winter; children were possibly more likely to be close to outdoor smokers in warmer temperatures. This also raises the question of whether children are more exposed to SHS in outdoor locations that are frequented more often in warmer temperatures, such as playgrounds, parks, sports fields and other places where children and parents gather.

Our findings are not consistent with a study from Milan that found a 78% increase in cotinine urine values in the winter compared to the spring, suggesting higher exposure of children in colder weather in that study [18]. It may be more challenging to smoke outdoors, or to smoke at an indoor location away from children such as a garage, in a large European center where apartments are more common than the single homes with attached or detached private garages on the property, typical in our location.

The number of our participants above the CCR cutoff under the age of 1 was greater than other age groups, although the average smokers per home were comparable across all age groups. This suggests that a younger child may have more exposure, potentially due to decreased mobility or increased in-person adult supervision or be relatively more susceptible to uptake or persistent cotinine levels due to their smaller mass, differences in substance clearance and increased metabolic demand; THS may also play a role due to younger children’s positioning and behavior [1,6]. This is congruent with the findings of a Chinese cohort study which suggests that younger populations have a higher vulnerability to SHS exposure [19]. The same level of exposure may have less impact on older children due to a different rate of uptake and nicotine clearance, and the ability to move away from a smoking adult, as well as potentially reflecting decreased direct parental supervision as children get older [19].

This study faced multiple limitations. Questionnaires are vulnerable to reporting bias, particularly as smoking is a stigmatized behavior, particularly parental smoking around children instead of confining this exposure outdoors. This self-reporting bias may explain why more winter participants did not report a smoking location. While the study attempted to address reporting bias through participant anonymity, it may still have impacted responses. Smokers may also have been less likely to choose to participate, although the proportion of smokers in the study was similar to the proportion of people who report smoking provincially, suggesting that the study population was representative of the overall population [20]. There were incomplete data; 14 participants indicated smoking status without reporting smoking location. This was a sample of convenience, and children provided urine samples at the time of their pediatric clinic appointment, which limited the number of samples collected and did not allow for the same caregiver-child dyads to be followed in summer and winter.

## 5. Conclusions

In our Canadian study, smoking location within the home, seasonality and temperature had no significant impact on urine cotinine, a measure of SHS exposure, in children aged 0–9. However, a child’s age and the number of cigarettes smoked per household did significantly impact measurable SHS exposure in children. Reported smoking behaviors did differ between summer and winter, with more smokers reporting indoor smoking in the wintertime than the summer, although numbers did not reach significance. A reasonable proportion of self-reported smoking families in our survey limited indoor smoking to the garage. Follow-up studies are needed to delineate if isolated indoor locations, such as an attached or detached garage, might be protective for children. Physicians should consider recommending smoking outdoors only, or limiting indoor smoking to a closed, segregated area such as a garage for those unwilling to smoke outdoors, as a harm mitigation strategy to prevent detectable SHS exposure. Although evidence is limited, this could also limit THS exposure. Other factors associated with the smoking location, such as attachment or shared ventilation with the home, proximity to children’s play areas, schools and other gathering areas should be considered. Providing information and support to smokers in non-stigmatizing ways during child health visits can help protect children, particularly infants, from SHS and THS exposure by limiting the number of cigarettes smoked and isolating smoking away from children.

## Figures and Tables

**Table 1 ijerph-19-07076-t001:** Seasonality of smoking for all participants who completed questionnaires.

Season	Winter	Summer	*p* Value
Number of participants	100	133	
Number of participants in smoking homes (%)	27 (27)	23 (17.3)	0.074
Location of Smoking			0.027
No location indicated (%)	11/27 (40.7)	3/23 (13.0)	
Outdoor Only (%)	5/27 (18.5)	3/23 (13.0)	
Garage (%)	4/27 (14.8)	12/23 (52.2)	
Indoor (%)	7/27 (25.9)	5/23 (21.7)	
Mean number of smokers per smoking home (SD)	2.6 (2.4)	1.3 (0.5)	0.015

**Table 2 ijerph-19-07076-t002:** CCR ratios of patients by age.

Age (Years)	<1	1–4	5–9	*p* Value
Number of participants	30	49	52	
Number of participants in smoking homes (%)	11 (36.7)	13 (26.5)	9 (17.6)	0.145
Mean number of smokers per smoking home (SD)	1.4 (0.7)	1.8 (1.5)	2.4 (2.1)	0.286
Participants above 30 ng/mg CCR cutoff (%)	11 (36.7)	4 (8.2)	3 (5.9)	<0.001
Mean CCR for participants above cutoff (SD)	832.7 (997.6)	507.2 (768.8)	357.2 (423.6)	0.812
Median CCR median (IQR)	2.02 (0.00 to 110.68)	2.17 (0.00 to 10.14)	1.57 (0.01 to 4.84)	

CCR = cotinine:creatinine ratio.

**Table 3 ijerph-19-07076-t003:** Seasonality of smoking for participant dyads who completed questionnaires and provided a child urine sample.

Season	Winter	Summer	*p* Value
Number of participants	62	71	
Number of participants in smoking homes (%)	17 (27.4)	20 (28.2)	0.923
Location of Smoking			0.037
No location indicated (%)	7/17 (41.2)	4/20 (20.0)	
Outdoor Only (%)	3/17 (17.6)	8/20 (40.0)	
Garage (%)	1/17 (5.9)	6/20 (30.0)	
Indoor (%)	6/17 (35.3)	2/20 (10.0)	
Mean number of smokers per smoking home (SD)	2.4 (1.9SD)	1.2 (0.4SD)	0.020
Participants above 30 ng/mg CCR cutoff (%)	10 (16.1)	8 (11.3)	0.414
Mean CCR for participants above cutoff (SD)	585.5 (961.4)	97.1 (176.5)	0.084
Median CCR median (IQR)	1.04 (0.00 to 12.14)	2.03 (0.00 to 5.89)	

CCR = cotinine:creatinine ratio.

**Table 4 ijerph-19-07076-t004:** Bivariate analysis of associations between tobacco exposure questionnaire factors and meeting the CCR cutoff.

Questionnaire Factors	*p* Value
**Household Smoking Information**	
Number of smokers in the home	<0.001
Number of cigarettes smoked in the household	<0.001
**Smoking Location**	
Television	0.240
Window	0.002
Kitchen	0.240
Near Ventilation	0.240
Garage	0.024
Outdoors	<0.001
Car	0.220

## Data Availability

The data presented in this study are available on request from the corresponding author.

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
