# Peer review of "Unintentional Tobacco Smoke Exposure in Children"

_ijerph, 2022, doi:10.3390/ijerph19127076_

Round 1
Reviewer 1 Report
This is an interesting and important paper, well written and presented. I have the following suggestions:
Add significance statistics to Tables 1,2,3 S1, S2
Page 4, report the multivariate analysis in an additional table in text (not supplement)
Authors should include what follow up studies are warranted based on these results
Authors should include what are the recommendations for families with smokers, and for physicians caring for families who have smokers.
Author Response
Reviewer 1
This is an interesting and important paper, well written and presented.
Thank you.
I have the following suggestions:
Add significance statistics to Tables 1,2,3 S1, S2
Thank you for this suggestion. We have made this change.
Page 4, report the multivariate analysis in an additional table in text (not supplement)
We have added Table 4, entitled “Bivariate analysis of associations between tobacco exposure questionnaire factors and meeting the CCR cutoff”, to the manuscript and removed Table S3 from the supplementary materials.
Authors should include what follow up studies are warranted based on these results
Thank you for this suggestion. We have added the following to our conclusions: “Follow-up studies are needed to delineate if isolated indoor locations, such as an attached or detached garage, might be protective for children and can be used as a harm mitigation strategy to prevent detectable SHS exposure.”
Authors should include what are the recommendations for families with smokers, and for physicians caring for families who have smokers.
We agree that it is important to offer translational content for these results and have added the following to our manuscript: “Physicians should consider recommending smoking outdoors only, or consideration of limiting indoor smoking to a closed, segregated area like a garage for those unwilling to smoke outdoors, as a harm mitigation strategy to prevent detectable SHS exposure.”
Reviewer 2 Report
Brief Summary: This paper reports the results of a cross-sectional study of the relationship between self-reported caregiver smoking, location, outdoor temperature, and children's urine cotinine as a marker for secondhand smoke (SHS) exposure. The methods include a self-report questionnaire delivered to child caregivers recruited from a child health clinic in Alberta, Canada. Urine was collected from the children (aged 0-9) and tested for cotinine/creatine. This paper adds to the body of knowledge on SHS exposure in children. It's main strengths are the combined self-report and urine cotinine data, and the inclusion of self-reported locations of smoking in the home, along with environmental factors such as temperature and seasonality.
General Concept Comments: While overall, this is a novel and interesting paper that will be useful for public health, there are some areas of weakness. In the background section and discussion, thirdhand smoke (THS) is not addressed (i.e., residual nicotine and other chemicals left on indoor surfaces by tobacco smoke). THS builds up over time and can be in the environment long after smoking has stopped. Research has shown that infants and young children may have increased exposure to THS due to their tendency to mouth objects and touch contaminated surfaces. Thus, in addition to active SHS exposure, THS may also contribute to urine cotinine. It is not clear whether or how this was considered in the study design. THS should be addressed in the Introduction and Discussion as a potential source of exposure.
Another area of weakness is the lack of detail in the methods. I understand that word limits may prevent lengthy descriptions of methods, but there are gaps regarding the collection of temperature data, and urine collection and analysis. Also, the type of homes that the participants lived in should be addressed.
As noted in the introduction, people who live in MUH may experience tobacco smoke exposure from smoking that occurs inside the home as well as from smoking that occurs in other units, hallways, balconies, etc. It's not clear how many participants in this study lived in detached or multi-unit housing. This may make a difference in how participants are reporting their locations of smoking, e.g., garage. It would be important to know how many participants had access to a garage. Also, for participants living in a high rise apartment building, going outside is more challenging and you may find more smoking indoors or out a window or on a balcony.
Another issue is that the discussion does not address the possibility of tobacco smoke exposure in locations away from the home, such as in the home of a relative or other caregiver, or other outdoor location. Was there any data collected on other sources of exposure?
Specific Comments:
Abstract:
- A sentence should be included in the abstract to briefly describe the methods of data collection. This could go after the aim ("This prospective study...") and before the results ("Of the 233 participants...")
- Caveats should be included in the abstract and conclusion when stating that garages might be protective for children and provide a harm mitigation strategy. While this may be the case in some contexts, if the garage is attached to the home and shares ventilation, or a bedroom or playroom is above the garage, or the garage is used as a play space, this may not reduce harm, and could increase harm in some situations.
Introduction:
-The last sentence of 1st par states "Up to 40-70% of children are exposed to SHS [1,4,5]." I'm assuming that this range is derived from various studies referenced and not an exact number. Also there is no geographic location or timeframe specified- is this globally, for Canada, US... is it current? This range needs some context and should be clarified if it is an estimate. Also reference 5 is incomplete. Are there any Canadian or provincial data that can be used? For example, the Canadian Health Measures Survey (CHMS) estimated SHS exposure using both questionnaire and biomonitoring approaches (cotinine analysis) https://www150.statcan.gc.ca/n1/pub/82-003-x/2021002/article/00002-eng.htm. Based on questionnaire data, CHMS reports the prevalence of nonsmokers regularly exposed to SHS was 14% for children (ages 6 to 11).
As mentioned in my general comments, the concept of THS should be raised in the introduction and accounted for in the methods and discussion.
Par 2 of the Introduction states, "While few studies focus on the risk at the level of locations in the home, more studies have identified that outdoor exposures at home are more common than indoor exposure." There are no references cited for this statement. The next sentence about MUH includes 2 citations (6,7)- do these also apply to outdoor vs. indoor exposures?
Ref 6 is a study based in Northern Thailand, which has a very different physical environment than Canada (i.e., warm climate all year around, access to outdoor spaces, ventilation, HVAC systems, etc.), and thus may not be as relevant to this study. However, there was a recent publication by Parks et al, 2022, 'Assessing secondhand and thirdhand tobacco smoke exposure in Canadian infants using questionnaires, biomarkers, and machine learning' (https://pubmed.ncbi.nlm.nih.gov/34175887/) that may be useful here. This study found that 12% of the sample reported that someone had smoked in the home since the child’s birth, 11% of the sample reported that smoking occurred outside of the house, and 1.6% reported that smoking occurred near a window or in the garage. Only 0.5% reported that smoking occurred inside the home. This is also relevant to the discussion.
In the third par of the introduction, the use of cotinine as a marker for tobacco smoke exposure is explained. The last sentence states: "Creatinine can be used to account for urine concentration". A brief explanation is needed to clarify what "accounting for urine concentration" means.
2.2 Urine collection and analysis: Was the urine collected by the parents or a lab technician?
2.3 Questionnaires: Was information collected about the type of home, i.e., MUH (with or without a balcony), detached home (with or without a garage). Was information collected about exposures away from the home, i.e., exposure in the home of a relative or other caregiver? This context is important for interpretation of data. If it was not collected or insufficiently reported, then this should be addressed in the discussion.
Also, did the questionnaire ask about smoking in the home, garage or car when a child was not present, which could help with interpretation of possible THS exposure?
3.1 Participants: If data were collected on type of home and access to a garage on the property, then this should be reported in the results. For example, the significance of the numbers of people reporting smoking in a garage is different if only a small portion of the sample had access to a garage.
Also, in 3rd par of section 3.1, please confirm the Table numbering- it appears that Table 1 should actually be Table 3, and Table 3 should actually be Table 2.
How was temperature data collected or determined?
3.2 Environmental Factors:
It's not clear where the temperature data came from and what the ranges mean.
Re. "A significant number of winter participants did not report smoking location..."- Could this be a reflection of self-report bias, where participants do not want to report smoking inside their home in the winter due to stigma and fear of being judged poorly?
In the third par of section 3.2, "Seasonality (Tables 2,3)....": I think this should be Tables 1 and 2?
Discussion:
- The comparison with the northern Thailand study is problematic. It makes sense that outdoor locations for smoking would be high in Thailand where the climate is much warmer all year around. Also, as mentioned earlier, the layout and physical environment around the home may be very different compared to Canada with possibly more access to outdoors. This should be qualified in the text- as was done for the Milan study comparison in the following par.
-Comparison with the Parks et al. study mentioned in my earlier comments may also be relevant here as it has Canadian context.
- I think the interpretation that children may be more likely to be close to outdoor smokers in warmer temperatures is valid and relevant for policy. It raises the question of whether children might also be exposed in other outdoor locations (park or playground; outside shops, arenas, sports fields; sidewalks, bus stops, etc.), and the potential for cumulative exposures to tobacco smoke.
- At the end of par 2, it is explained that single homes with attached or detached private garages on the property were typical in your study. This is important to raise earlier in the paper as it is critical to interpretation of the data (re. see my earlier comments).
- Par 3: It is explained that "a younger child may have more exposure, potentially due to decreased mobility, increased in-person adult supervision, or be relatively more susceptible to uptake or persistent cotinine levels due to their smaller mass, differences in substance clearance and increased metabolic demand." This is an important recognition and needs supporting reference(s). This also is closely tied to the concept of THS as younger children are more likely to be in contact with contaminated surfaces.
- Par 4: The discussion of reporting bias is very important and could be emphasized more that stigma regarding smoking is particularly strong with respect to parents/guardians smoking around their children. There may also be fear of being reported to CAS, etc.
Conclusion:
-"In our Canadian study...temperature [HAD] no significant impact..."
- While I agree that it would be interesting to conduct further research to delineate whether an attached or detached garage might be used as a harm reduction strategy; as mentioned in my earlier comments, I think this depends on circumstances, i.e., if attached to the home, what is the ventilation, is there a room above the garage; is the garage used as a play space, etc.
-I think what is missing here is the potential for counselling and education for parents/guardians about SHS exposure, and support for reducing and quitting smoking. This is mentioned in the abstract but not in the conclusions of the paper. Proving information and support in non-stigmatizing ways in locations where there is frequent contact with parents, such as child health clinics could be one approach.
Author Response
Reviewer 2
Brief Summary: This paper reports the results of a cross-sectional study of the relationship between self-reported caregiver smoking, location, outdoor temperature, and children's urine cotinine as a marker for secondhand smoke (SHS) exposure. The methods include a self-report questionnaire delivered to child caregivers recruited from a child health clinic in Alberta, Canada. Urine was collected from the children (aged 0-9) and tested for cotinine/creatine. This paper adds to the body of knowledge on SHS exposure in children. It's main strengths are the combined self-report and urine cotinine data, and the inclusion of self-reported locations of smoking in the home, along with environmental factors such as temperature and seasonality.
General Concept Comments: While overall, this is a novel and interesting paper that will be useful for public health, there are some areas of weakness. In the background section and discussion, thirdhand smoke (THS) is not addressed (i.e., residual nicotine and other chemicals left on indoor surfaces by tobacco smoke). THS builds up over time and can be in the environment long after smoking has stopped. Research has shown that infants and young children may have increased exposure to THS due to their tendency to mouth objects and touch contaminated surfaces. Thus, in addition to active SHS exposure, THS may also contribute to urine cotinine. It is not clear whether or how this was considered in the study design. THS should be addressed in the Introduction and Discussion as a potential source of exposure.
Thank you for your evaluation of our study. In order to address the potential contribution of third hand smoke we have made the following changes:
- Introduction P1: “Potential exposures to thirdhand smoke (THS), the residual chemicals left on surfaces following smoking, present another route to indirect pediatric exposure “
- Discussion P2: “While outdoors was found to be the most common location of smoking and is associated with higher exposure to SHS, THS should also be considered, especially in households with indoor smokers. In a 2004 study addressing thirdhand smoke, urine cotinine levels were significantly higher in infants who lived in households with indoor smokers compared to households with outdoor smokers. There is a lack of public awareness and understanding of THS exposure; with 85% of tobacco smoke not being odorous or visible leading to to a gap in knowledge among the public.”
- Discussion P4: THS may also play a role due to younger children’s positioning and behavior
Another area of weakness is the lack of detail in the methods. I understand that word limits may prevent lengthy descriptions of methods, but there are gaps regarding the collection of temperature data, and urine collection and analysis. Also, the type of homes that the participants lived in should be addressed.
Thank you for this constructive feedback regarding our study. In order to provide more detail in the methods, we elaborated on our urine collection and analysis methods and included a separate header regarding temperature and season. Section 2.2 in the updated manuscript describes urine collection and Section 2.4 the season and temperature.
The participants’ homes have also been described (Results, P1), as well as additional regular exposures to SHS such as in the homes of relatives.
As noted in the introduction, people who live in MUH may experience tobacco smoke exposure from smoking that occurs inside the home as well as from smoking that occurs in other units, hallways, balconies, etc. It's not clear how many participants in this study lived in detached or multi-unit housing. This may make a difference in how participants are reporting their locations of smoking, e.g., garage. It would be important to know how many participants had access to a garage. Also, for participants living in a high rise apartment building, going outside is more challenging and you may find more smoking indoors or out a window or on a balcony.
Thank you for this feedback. Although we did not collect information on how many participants had access to a garage, we added data about how many participants reside in single family housing versus multiunit homes (Results P2). “The 209 (90%) participants who completed questionnaires reported residing in single detached homes.”
Another issue is that the discussion does not address the possibility of tobacco smoke exposure in locations away from the home, such as in the home of a relative or other caregiver, or other outdoor location. Was there any data collected on other sources of exposure?
Thank you for this feedback. Data was collected on other sources of exposure, and we updated our results to reflect this (Results P2) by adding: “Smoke outside of the home at least once a week was reported by 29 (12%) of participants, from locations including public areas, the homes of relatives, neighbours, and friends.”
Specific Comments:
Abstract:
- A sentence should be included in the abstract to briefly describe the methods of data collection. This could go after the aim ("This prospective study...") and before the results ("Of the 233 participants...")
Thank you for this suggestion. To address this gap in our abstract, we added the following: “Participants aged 0-9 were recruited from Child Health Clinics at the Misericordia Community Hospital in Edmonton, Alberta between January 8 to February 24, 2016 and June 30th to August 18, 2016. Participant CCR levels were compared to caregiver-reported smoking location and environmental factors like temperature and season.”
Caveats should be included in the abstract and conclusion when stating that garages might be protective for children and provide a harm mitigation strategy. While this may be the case in some contexts, if the garage is attached to the home and shares ventilation, or a bedroom or playroom is above the garage, or the garage is used as a play space, this may not reduce harm, and could increase harm in some situations.
Thank you for this suggestion. We made the following change to the abstract: “Segregated areas such as a garage may provide a useful harm mitigation strategy for indoor smokers, provided the garage does not share ventilation or is not in close proximity to high-traffic areas of the home”. We also added the following to the conclusion: “Follow-up studies are needed to delineate if isolated indoor locations, such as an attached or detached garage, might be protective for children and can be used as a harm mitigation strategy to prevent detectable SHS exposure. These should also factor in circumstances surrounding the smoking location, such as if it is attached to the home, shares ventilation with the home, is in close proximity to children’s play areas.”
Introduction:
The last sentence of 1st par states "Up to 40-70% of children are exposed to SHS [1,4,5]." I'm assuming that this range is derived from various studies referenced and not an exact number. Also there is no geographic location or time frame specified- is this globally, for Canada, US... is it current? This range needs some context and should be clarified if it is an estimate. Also reference 5 is incomplete. Are there any Canadian or provincial data that can be used? For example, the Canadian Health Measures Survey (CHMS) estimated SHS exposure using both questionnaire and biomonitoring approaches (cotinine analysis) https://www150.statcan.gc.ca/n1/pub/82-003-x/2021002/article/00002-eng.htm. Based on questionnaire data, CHMS reports the prevalence of nonsmokers regularly exposed to SHS was 14% for children (ages 6 to 11).
Thank you for this suggestion. We have made the following changes: The 40-70% range was from a single source and the additional citations were removed. We have also added "Up to 40-70% of children are exposed to SHS globally and approximately 14% of children are exposed to SHS in Canada [4,5]."
As mentioned in my general comments, the concept of THS should be raised in the introduction and accounted for in the methods and discussion.
Thank you for this suggestion. We have made the following addition to paragraph 1 of the introduction: “Potential exposures to thirdhand smoke, the residual chemicals left on surfaces following smoking, present another route to indirect pediatric exposure.”
Par 2 of the Introduction states, "While few studies focus on the risk at the level of locations in the home, more studies have identified that outdoor exposures at home are more common than indoor exposure." There are no references cited for this statement. The next sentence about MUH includes 2 citations (6,7)- do these also apply to outdoor vs. indoor exposures?
Thank you for this suggestion. We added two references to the statement: “While few studies focus on risk at the level of locations in the home, more studies have identified that outdoor exposures at home are more common than indoor exposure [7,8].”
Ref 6 is a study based in Northern Thailand, which has a very different physical environment than Canada (i.e., warm climate all year around, access to outdoor spaces, ventilation, HVAC systems, etc.), and thus may not be as relevant to this study. However, there was a recent publication by Parks et al, 2022, 'Assessing secondhand and thirdhand tobacco smoke exposure in Canadian infants using questionnaires, biomarkers, and machine learning' (https://pubmed.ncbi.nlm.nih.gov/34175887/) that may be useful here. This study found that 12% of the sample reported that someone had smoked in the home since the child’s birth, 11% of the sample reported that smoking occurred outside of the house, and 1.6% reported that smoking occurred near a window or in the garage. Only 0.5% reported that smoking occurred inside the home. This is also relevant to the discussion.
Thank you for this suggestion. We have addressed the comment in the discussion as follows: “In a recent Canadian study, it was reported that among the 12% of households that reported smoking at home after a child’s birth, 11% smoked outside of the house, 1.6% smoked near a window or in the garage, and only 0.5% smoked inside the home [8].”
In the third part of the introduction, the use of cotinine as a marker for tobacco smoke exposure is explained. The last sentence states: "Creatinine can be used to account for urine concentration". A brief explanation is needed to clarify what "accounting for urine concentration" means.
Thank you for this suggestion. We have changed the sentence to “Urine cotinine:creatinine ratio (CCR) can be used to address potential discrepancies due to inconsistent urine concentrations [13].”
2.2 Urine collection and analysis: Was the urine collected by the parents or a lab technician?
Thank you for this suggestion. We made the following change to 2.2 Urine Collection and Analysis: “For older children, urine was collected by parents in laboratory specimen jars. For infants, cotton balls in the infant’s diaper were used to collect urine then centrifuged to collect the urine from the cotton balls.”
2.3 Questionnaires: Was information collected about the type of home, i.e., MUH (with or without a balcony), detached home (with or without a garage). Was information collected about exposures away from the home, i.e., exposure in the home of a relative or other caregiver? This context is important for interpretation of data. If it was not collected or insufficiently reported, then this should be addressed in the discussion.
Thank you for this suggestion. We added the following changes to 3.1 Results: “The 209 (90%) participants who completed questionnaires reported residing in single detached housing.” and “The 115 (86%) participants who provided urine samples and completed questionnaires resided in single detached housing.” We also added data regarding exposure outside of the home to 3.1 Results: “29 (12%) of participants also reported exposure to smoke outside of the home at least once a week, from locations such as public areas, the homes of relatives, neighbours, and friends.”
Also, did the questionnaire ask about smoking in the home, garage or car when a child was not present, which could help with interpretation of possible THS exposure?
Thank you for this suggestion. We made the following changes to 2.3 Questionnaires: “The location of children during smoking was not recorded.”
3.1 Participants: If data were collected on type of home and access to a garage on the property, then this should be reported in the results. For example, the significance of the numbers of people reporting smoking in a garage is different if only a small portion of the sample had access to a garage.
Thank you for this suggestion. Unfortunately, the presence of a garage on the property was not recorded.
Also, in 3rd par of section 3.1, please confirm the Table numbering- it appears that Table 1 should actually be Table 3, and Table 3 should actually be Table 2.
Thank you for this suggestion. We have noted this mistake and changed the table numbers accordingly.
How was temperature data collected or determined?
Thank you for this suggestion. We added in clarification about how temperature data was collected: “Temperature was recorded daily by the recruiter by taking the midday outdoor temperature of the Edmonton area.”
3.2 Environmental Factors:
It's not clear where the temperature data came from and what the ranges mean.
Thank you for this suggestion. We made the following change: “For the 233 participants who completed questionnaires, temperature data was missing for 15 participants; the other 218 were distributed within temperature quartiles based on the highest and lowest recorded temperatures during the recruitment period:”
Re. "A significant number of winter participants did not report smoking location..."- Could this be a reflection of self-report bias, where participants do not want to report smoking inside their home in the winter due to stigma and fear of being judged poorly?
Thank you for this suggestion. We addressed the potential for self-reporting bias in the winter in the Discussion: “Parental smoking, particularly around children, is further stigmatized. This self-reporting bias may explain why a significant number of winter participants did not report a smoking location.”
In the third par of section 3.2, "Seasonality (Tables 2,3)....": I think this should be Tables 1 and 2?
Thank you for this suggestion. We have noted this mistake and changed the table numbers accordingly.
Discussion:
The comparison with the northern Thailand study is problematic. It makes sense that outdoor locations for smoking would be high in Thailand where the climate is much warmer all year around. Also, as mentioned earlier, the layout and physical environment around the home may be very different compared to Canada with possibly more access to outdoors. This should be qualified in the text- as was done for the Milan study comparison in the following par.
Comparison with the Parks et al. study mentioned in my earlier comments may also be relevant here as it has Canadian context.
Thank you for this suggestion. We have made the following changes: “Smoking outside may be more convenient in Thailand than in Canada because of its warmer climate. Despite this difference in climate, exposure to SHS is also higher outdoors than indoors in Canada. In a recent Canadian study, it was reported that among the 12% of households that admitted to smoking after a child’s birth, 11% smoked outside of the house, 1.6% smoked near a window or in the garage, and only 0.5% smoked inside the home.”
I think the interpretation that children may be more likely to be close to outdoor smokers in warmer temperatures is valid and relevant for policy. It raises the question of whether children might also be exposed in other outdoor locations (park or playground; outside shops, arenas, sports fields; sidewalks, bus stops, etc.), and the potential for cumulative exposures to tobacco smoke.
Thank you for this point. We have added the following sentence to incorporate some of these potential exposures: “This also raises the question of whether children are more exposed to SHS in outdoor locations that are frequented more in warmer temperatures, such as playgrounds, parks, sports fields, and other places where parents and children gather.”
At the end of par 3, it is explained that single homes with attached or detached private garages on the property were typical in your study. This is important to raise earlier in the paper as it is critical to interpretation of the data (re. see my earlier comments).
Thank you for this suggestion. We have added the following changes earlier in the manuscript to 3.1 Results: “The 209 (90%) participants who completed questionnaires reported residing in single detached housing.” and “115 (86%) participants who provided urine samples and completed questionnaires resided in single detached housing.”
Par 3: It is explained that "a younger child may have more exposure, potentially due to decreased mobility, increased in-person adult supervision, or be relatively more susceptible to uptake or persistent cotinine levels due to their smaller mass, differences in substance clearance and increased metabolic demand." This is an important recognition and needs supporting reference(s). This also is closely tied to the concept of THS as younger children are more likely to be in contact with contaminated surfaces.
Thank you for this point. We have added two references to support our sentence and referred to a younger child’s potential for more exposure to THS.
Par 4: The discussion of reporting bias is very important and could be emphasized more that stigma regarding smoking is particularly strong with respect to parents/guardians smoking around their children. There may also be fear of being reported to CAS, etc.
Thank you for this suggestion. We have made the following changes: “Parental smoking, particularly around children, is further stigmatized leading to parents potentially being afraid of facing repercussions if smoking habits are reported accurately.”
Conclusion:
"In our Canadian study...temperature [HAD] no significant impact..."
Thank you for this suggestion. We changed the sentence to “...had no significant impact” to address this error.
While I agree that it would be interesting to conduct further research to delineate whether an attached or detached garage might be used as a harm reduction strategy; as mentioned in my earlier comments, I think this depends on circumstances, i.e., if attached to the home, what is the ventilation, is there a room above the garage; is the garage used as a play space, etc.
Thank you for this suggestion. We made the following change to the conclusion: “Follow-up studies are needed to delineate if isolated indoor locations, such as an attached or detached garage, might be protective for children and can be used as a harm mitigation strategy to prevent detectable SHS exposure. These should also factor in circumstances surrounding the smoking location, such as if it is attached to or shares ventilation with the home, or is in close proximity to children’s play areas."
I think what is missing here is the potential for counselling and education for parents/guardians about SHS exposure, and support for reducing and quitting smoking. This is mentioned in the abstract but not in the conclusions of the paper. Proving information and support in non-stigmatizing ways in locations where there is frequent contact with parents, such as child health clinics could be one approach.
Thank you for this suggestion. We made the following addition to the conclusion: “Providing information and support to smokers in non-stigmatizing ways during child health visitations can help to protect children, particularly infants, from SHS exposure by limiting the number of cigarettes smoked and isolating smoking to outside the home. Although evidence is limited this could also limit THS exposure.”